# Apolipoprotein and LRP1-Based Peptides as New Therapeutic Tools in Atherosclerosis

**DOI:** 10.3390/jcm10163571

**Published:** 2021-08-13

**Authors:** Aleyda Benitez Amaro, Angels Solanelles Curco, Eduardo Garcia, Josep Julve, Jose Rives, Sonia Benitez, Vicenta Llorente Cortes

**Affiliations:** 1Institute of Biomedical Research of Barcelona (IIBB), Spanish National Research Council (CSIC), 08036 Barcelona, Spain; Aleyda.benitez.Amaro@iibb.csic.es (A.B.A.); EgarciaR@santpau.cat (E.G.); 2Biomedical Research Institute Sant Pau (IIB-Sant Pau), 08041 Barcelona, Spain; angelssocu@gmail.com; 3Metabolic Basis of Cardiovascular Risk Group, Biomedical Research Institute Sant Pau (IIB Sant Pau), 08041 Barcelona, Spain; jjulve@santpau.cat; 4CIBER de Diabetes y Enfermedades Metabólicas Asociadas (CIBERDEM), 28029 Madrid, Spain; 5Biochemistry Department, Hospital de la Santa Creu i Sant Pau, 08025 Barcelona, Spain; JRives@santpau.cat; 6Department of Biochemistry and Molecular Biology, Faculty of Medicine, Universitat Autònoma de Barcelona (UAB), Cerdanyola del Vallès, 08016 Barcelona, Spain; 7Cardiovascular Biochemistry Group, Biomedical Research Institute Sant Pau (IIB Sant Pau), 08041 Barcelona, Spain; 8CIBERCV, Institute of Health Carlos III, 28029 Madrid, Spain

**Keywords:** apolipoproteins, atherosclerosis, clinical trials, LDL, peptides

## Abstract

Apolipoprotein (Apo)-based mimetic peptides have been shown to reduce atherosclerosis. Most of the ApoC-II and ApoE mimetics exert anti-atherosclerotic effects by improving lipid profile. ApoC-II mimetics reverse hypertriglyceridemia and ApoE-based peptides such as Ac-hE18A-NH2 reduce cholesterol and triglyceride (TG) levels in humans. Conversely, other classes of ApoE and ApoA-I mimetic peptides and, more recently, ApoJ and LRP1-based peptides, exhibit several anti-atherosclerotic actions in experimental models without influencing lipoprotein profile. These other mimetic peptides display at least one atheroprotective mechanism such as providing LDL stability against mechanical modification or conferring protection against the action of lipolytic enzymes inducing LDL aggregation in the arterial intima. Other anti-atherosclerotic effects exerted by these peptides also include protection against foam cell formation and inflammation, and induction of reverse cholesterol transport. Although the underlying mechanisms of action are still poorly described, the recent findings suggest that these mimetics could confer atheroprotection by favorably influencing lipoprotein function rather than lipoprotein levels. Despite the promising results obtained with peptide mimetics, the assessment of their stability, atheroprotective efficacy and tissue targeted delivery are issues currently under progress.

## 1. Introduction

In the past decades, worsening of eating habits and sedentary lifestyle factors have increased worldwide incidence of pathologies associated with cardiovascular risk. Nowadays, cardiovascular diseases cause more than 17.5 million deaths annually, representing an enormous health and socioeconomic problem. Moreover, population aging and gradual accumulation of risk factors (diabetes, dyslipidemia, obesity, and hypertension) are key determinants of the growth of cardiovascular disease prevalence. One of the main underlying causes of cardiovascular diseases is atherosclerosis. Atherosclerotic lesions are strongly and uninterruptedly promoted by cholesteryl ester-enriched lipoproteins, mainly lipoprotein remnants and low-density lipoproteins (LDL). Modified lipoproteins accumulate in the arterial wall and exert locally proatherogenic actions, including the induction of intracellular lipid accumulation and inflammation [1]. Aggregated LDL (agLDL) is one of the main LDL modifications occurring in the arterial intima. The agLDL cause both extra- and intracellular cholesterol accumulation in the vascular wall [2,3].

There are several therapeutic strategies that combine healthy lifestyle with pharmacological interventions to modulate atherosclerosis through the control of lipoprotein metabolism. The most widely used drugs in primary and secondary prevention are statins or HMG-CoA reductase inhibitors. The inhibition of HMG-CoA reductase reduces intracellular cholesterol and increases LDL receptor levels in the liver, facilitating increased blood LDL clearance [4]. Despite their well-established efficacy in reducing blood LDL-cholesterol levels and high tolerability, prolonged use causes adverse effects such as intolerance, myalgia, and myopathy. Statin-treated patients could also experience higher incidence of diabetes [5] due to the interference of statins with insulin sensitivity and pancreatic β-cell functionality [6].

Statin treatment was not always associated to a reduction in cardiovascular mortality [7]. Other emergent hypolipemiant therapies, such as inhibitors of proprotein convertase subtilisin/kexin type 9 (PCSK9), either alone or in combination with statin treatment, have been recommended in patients with high cardiovascular risk [8]. The PCSK9 enzyme binds to the LDL receptor (LDLR), resulting in accelerated LDLR degradation and increased LDL-circulating cholesterol levels [9]. PCSK9 inhibitors are monoclonal antibodies that selectively bind to circulating PCSK9, preventing PCSK9 binding to the LDL receptor (LDLR) and PCSK9-mediated LDLR degradation. Patients with spontaneous functional PCSK9 deficiency showed lower levels of LDL cholesterol (LDL-C) and reduced long-term cardiovascular risk [10]. Through the reduction of LDL-C, TG, and very-low-density lipoprotein cholesterol (VLDL-C) and the increase in high-density lipoprotein (HDL) cholesterol (HDL-C) levels, these inhibitors efficiently protect against cardiovascular diseases [11].

Peptides and their mimetics with capacity to improve lipid profile and/or lipoprotein functionality have emerged as potential therapeutic tools in cardiometabolic diseases and particularly, in atherosclerosis. These peptides are usually designed as short amino acid chains based on functional domains of apoproteins or apoprotein receptors. Advanced technologies have helped to progressively improve the therapeutic characteristics of these peptides. There are several peptides that have already met a wide range of therapeutic objectives, including good affinity profiles, oral bioavailability, low toxicity, biosecurity, tolerability, good efficacy, high potency, and selectivity [12].

Recombinant human insulin (Humulin) was the first peptide produced for cardiometabolic therapy [13], in particular, insulin replacement. Humulin showed limited subcutaneous absorption due to its tendency to form complexes that limited its passive transport across the endothelium. Consequently, plasma humulin increased at a slower rate than endogenous insulin levels and had a substandard brief half-life. To overcome these drawbacks, insulin lispro, a human insulin analog, was engineered with an exchange of amino acid proline and lysine at positions 28 and 29 in the beta chain of insulin [14]. Other example of peptide designed for metabolic therapy was desmopressin, a vasopressin-based peptide with D chirality amino acids (AA). Desmopressin is currently used in the treatment of neurogenic diabetes insipidus, a disease caused by vasopressin deficiency. The introduction of D-AA in the sequence increases the peptide resistance against proteases and prolongs its half-life. Desmopressin can be administered nasally, intravenously, or sublingually and optimally reach the glycemic control of patients not responding to antidiabetic medication. Liraglutide, an analog of glucagon-like peptide-1 (GLP-1) that activates the GLP-1 receptor, is currently used for the treatment of type 2 diabetes.

In the vascular field, calcitonin gene-related peptide (CGRP) is a neuropeptide produced by calcitonin gene alternative RNA processing. CGRP is a potent vasodilator that protects the cardiovascular system and facilitates wound healing [15].

According to Transparency Market Research (TMR), the market value of cardiovascular drugs will exceed US$ 91 billion by 2025 [16]. The bulk of this market corresponds to peptide drugs that have gained ground in pharmaceutical research due to their low production costs.

In this review, we focus on mimetic peptides with anti-atherosclerotic properties, highlighting peptide-preventing effects on LDL instability and aggregation. LDL aggregation is a key determinant of LDL proatherogenicity. This review (1) updates in vitro and in vivo studies using mimetic peptides and their influence on atherosclerosis, (2) discusses the therapeutic potential of LRP1-derived peptides, and (3) summarizes clinical trials using apolipoprotein-based peptides.

## 2. Apolipoprotein and LRP1-Based Peptides. Structure and Protective Effects against LDL Modification

Apolipoprotein and LRP1-based peptides are short, chemically synthesized peptides that mimic amino acid sequences of natural apolipoproteins or lipoprotein receptors. Given their amphipathic nature, these peptides bind to lipoproteins, conferring structural and functional protection [17]. Usually, these peptides are synthesized using a D instead of L amino acid configuration to make them more resistant to protease-induced degradation. This section provides a summary of peptide sequences (Table 1), structure, and characteristics of the peptide-apoprotein interactions contributing to lipoprotein protection against modification.

### 2.1. ApoA-I-Based Peptides

Apolipoprotein A-I (ApoA-I), the main apolipoprotein in HDL, is essential for HDL generation and function. ApoA-I is composed of 243 amino acids structured in 10 amphipathic α-helices, most of them essential for lipid interchange [33]. The interaction of amphipathic α-helices of ApoA-I, either purified or in HDL particles, with LDL prevents LDL from non-enzymatic and enzymatic modifications [34,35,36]. The enzymatic modification of LDL by sphingomyelinase (SMase) and phospholipase C of the arterial intima plays a crucial role in LDL aggregation [37,38]. The ApoA-I anti-aggregative properties have been mainly explained by ApoA-I binding to LDL surface hydrophobic areas [36].

Peptides derived from ApoA-I, and particularly peptide 4F, have been shown to exert not only anti-LDL-aggregating effects, but an additional wide variety of anti-atherosclerotic effects [12,39]. 4F peptide is composed of 18 amino acids and four phenylalanine residues (F) that form an α-helical secondary structure on the hydrophobic side. The 4F peptide protects LDL against SMase-induced aggregation regardless of pH (neutral or acidic) or degree of LDL aggregation. The binding of 4F to LDL causes a rearrangement of lipids that stabilize the particle against potential ApoB-100 conformational changes [36].

### 2.2. ApoC-II-Based Peptides

Apolipoprotein C-II (ApoC-II) is an exchangeable protein composed of three amphipathic α-helices that are located at the surface of chylomicrons, HDL, and very-low-density lipoproteins (VLDL) lipoproteins. In the postprandial state, ApoC-II is mainly carried by VLDL and LDL [40]. ApoC-II has one lipid-binding domain located at the N-terminal end and one lipoprotein lipase (LPL) activation domain located at the C-terminal end. The hydrophobic hinge residues of the lipid-binding domain allow a temporal interaction of ApoC-II with the lipids of the lipoproteins surfaces, as well as their transfer between lipoproteins [41]. A novel human ApoC-II mimetic bihelical peptide, C-II-a has recently been developed. Peptide C-II-a is formed by an amphipathic helix that contains the amino acid sequence of 18A peptide (ApoA-I-based peptide that confers cholesterol efflux stimulating properties) and a motif based on the last helix of apolipoprotein C-II (that activates LpL-induced lipolysis) [42].

### 2.3. ApoE-Based Peptides

Apolipoprotein E (ApoE) is a multifunctional secreted glycoprotein of 299 amino acids mainly synthesized by the liver. ApoE consists of two domains separated by a hinge region: the LDLR-binding globular domain (1–191 residues) and the lipid-binding domain with an amphipathic α-helical motif (192–299 residues) [43]. The globular domain exerts anti-inflammatory actions and promotes the endocytic clearance of VLDL and remnant lipoproteins through the LDLR, while the lipid-binding domain promotes cholesterol efflux from macrophages [44]. In humans, there are three structural isoforms of the protein, E2, E3, and E4, resulting from cysteine-arginine interchanges at residues 112 and 158, with subsequent different functional properties and pathological consequences. ApoE3 mediates apolipoprotein-apolipoprotein interactions and shows a higher affinity for HDL [45]. In contrast, ApoE4 has a higher affinity for TG-enriched lipoproteins, such as VLDL, due to the rearrangement of its C-terminal domain, which enhances its capacity to interact with lipids present in VLDL but not in HDL. In vitro studies have demonstrated that purified ApoE can inhibit LDL aggregation induced by lipolytic enzymes, such as phospholipase C [35].

ApoE genetic variations are associated not only with plasma lipid levels but also with atherosclerosis risk and pathobiology of neurodegenerative diseases [46,47].

ApoE peptides have been designed to contain key functional sequences of the LDLR-binding globular domain of ApoE. One of the most promising peptides is Ac-hE18A-NH_2_, a dual-domain cationic apolipoprotein-mimetic peptide composed of the LDLR-binding region of ApoE (141–150 residues), and the peptide 18A (mimics the lipid-binding region of ApoA-I [24]. The peptide mR18L is a single domain cationic amphipathic helical peptide derived from the lytic class L peptide 18L modified by incorporating aromatic residues in the nonpolar face [28]. EpK peptide, which contains an LDLR-binding region and a lipid-binding region linked by six Lysine residues, has also shown protective anti-atherosclerotic effects [26]. hEp was designed based on the structure and function of EpK and contains nearly the entire amphipathic helix 4 of the N-terminal domain of the LDLR binding region and the major C-terminal lipid-binding region of ApoE. hEp peptide injection could immediately lower plasma cholesterol level within 4 h in mice by exhibiting both lipid-binding and LDLR-binding activity [27]. ApoEdp is a tandem repeat dimer peptide of ApoE amino acids (141–149) that adopts an α-helical structure and mimics the LRP1-binding region of ApoE. It is more stable than the monomeric version and has been proposed for the treatment of diabetic retinopathy [30].

### 2.4. ApoJ-Based Peptides

Apolipoprotein J (ApoJ), also referred as clusterin is a 449 amino acid protein that acts extra- or intracellularly as a chaperone. The mature protein is a glycosylated heterodimer formed by α- and β-chains linked through five disulfide bonds. Approximately 20.5% of circulating ApoJ is associated with lipoproteins (18.5% HDL, 0.9% LDL, and 1.1% VLDL), although the content of lipoprotein-associated ApoJ decreases in hyperlipidemia [48]. In HDL, ApoJ is carried by a subset of dense particles containing ApoA-I and paraoxonase (PON) [49]. ApoJ may also be associated with LDL. Particularly, ApoJ has been identified in one of the modified forms of circulating LDL, an electronegative LDL subfraction (LDL (-)) [50], with inflammatory properties, which is relatively increased in high-risk cardiovascular patients [51].

Navab et al. synthesized seven peptides corresponding to seven G1 amphipathic helices of apoJ. Among them, the active peptides correspond to 113–122 residues, which were synthesized as D-amino acids [31]. D-(113–122) ApoJ, a 10-residue peptide spanning the predicted class G amphipathic helix 6 from ApoJ, prevented SMase-induced LDL aggregation through its ability to bind to hydrophobic regions of LDL particle surface prone to LDL aggregation [52,53].

### 2.5. LRP1-Based Peptides

LRP1 is a heterodimeric cell surface receptor belonging to the LDL receptor family involved in several biological processes and signaling pathways. In human coronary vascular smooth muscle cells (hcVSMCs), LRP1 is the main receptor responsible for the uptake of esterified cholesterol from agLDL [54]. Hypercholesterolemia upregulates LRP1 expression in hcVSMC and macrophages causing foam cell formation [55,56,57,58]. LRP1 interacts with agLDL through the CR9 domain located in cluster II of the LRP1 receptor. In particular, the sequence Gly^1127^-Cys^1140^, which covers the C-terminal half of the CR9 domain, is crucial for agLDL interaction and the cellular uptake of cholesteryl esters from agLDL [58]. From the sequence Gly^1127^-Cys^1140^, the LP3 peptide and other stable retroenantiomer peptides, such as DP3, were structurally optimized to achieve maximal functionality [32,59].

Biochemical studies demonstrated that both LP3 and DP3 peptides prevent SMase- and phospholipase A2 (PLA2)-induced LDL aggregation by binding to a specific sequence of ApoB-100 [32]. LDL proteomics and computational modeling methods demonstrated that these LRP1-based peptides preserve ApoB-100 conformation due to their electrostatic interaction with a basic region of ApoB-100. The peptide–apolipoprotein interaction is determined by the formation of salt bridge contacts between two of the acidic residues in DP3 (namely, D-Glu^3^ and D-Glu^9^) and two positively charged ApoB-100 residues (Lys^3229^ and Lys^3234^) [32].

A new family of LRP1-P3 peptide derivatives with enhanced potency and proteolytic stability were designed through in silico conformational sampling and ApoB-100 molecular docking. These peptides (a total of 46) were tested using a dual (biochemical-cellular) screening assay. This new family of peptides contains linear, fragment, cyclic, and alanine scanning derivatives and has been generated through two consecutive optimization rounds. Structurally and functionally optimized peptides contain hotspot residues that were replaced by alanine, because this strategy confers an increased capacity to form prone α-helix conformations that are crucial for electrostatic interaction with ApoB-100. These new compounds were proven to efficiently inhibit LDL aggregation promoted by SMase and PLA2 [59].

## 3. Impact of Peptides on Atherosclerosis

In this section, we review the mechanisms underlying the effects of these peptides in atherosclerosis.

### 3.1. ApoA-I-Based Peptides

Both L-4F and D-4F peptides reduce atherosclerotic plaque burden, pro-inflammatory cytokine secretion, insulin resistance, and lipid oxidation in biochemical, cellular, and in vivo models [60,61]. A different mimetic, the 37pA, exhibits a higher capacity than 4F to increase reverse cholesterol transport (RCT) both dependent and independently of ATP-binding cassette subfamily A member 1 (ABCA1) in cell culture experiments [19]. In contrast to other ApoA-I mimetic peptides, ELK-2A2K2E significantly reduces TGs in the plasma of ApoE-deficient mice [21]. This peptide also reduces the production of pro-inflammatory cytokines (TNF-α and IL-6) by cholesterol-loaded macrophages [62].

### 3.2. ApoC-II-Based Peptides

The 18A-C-II-a peptide enhances lipolysis in plasma from patients with ApoC-II deficiency and with other forms of hypertriglyceridemia in ex vivo studies [63]. This peptide also stimulates LPL activity in isolated lipoproteins [42]. In addition, 18A-C-II-a peptide strongly reduced plasma TG levels in different animal models, including apoC-II-KO zebrafish [64], ApoE-deficient mice [22], and ApoC-II-mutant mice [65]. In the latter model, as well as in wild-type mice, 18A-C-II-a peptide promotes the clearance of TG-rich lipid emulsions from blood and the transfer of fatty acids (FA) to specific peripheral tissues with FA oxidative capacity [66]. D6PV peptide strongly decreases TGs in human plasma by exerting a dual-action, activation of LpL, and inhibition of apoC-III [23]. Noteworthy, this peptide activates LpL in the plasma of hypertriglyceridemic patients more efficiently than the native form of ApoC-II. D6PV also reduces plasma TG, ApoC-III, and ApoB levels in humanized ApoC3 mice [23]. Taken together, these results suggest that D6PV could be an efficient treatment for hypertriglyceridemia.

### 3.3. ApoE-Based Peptides

ApoE has protective effects on the progression of atherosclerosis by promoting the hepatic uptake of atherogenic lipoproteins and macrophage cholesterol efflux [67]. Consistently, the deficiency of ApoE in mice develops hypercholesterolemia, inflammation, and atherosclerosis, whereas apoE expression protects the mice against this phenotype [68,69]. ApoE-derived mimetic peptides have been proposed as candidates for anti-atherogenic therapy [70]. The chimeric protein Ac-hE18A-NH_2_ reduces plasma cholesterol, promotes macrophage cholesterol efflux, shows anti-inflammatory and anti-oxidative properties, and reduces atherosclerotic lesion formation to a greater extent than 4F in ApoE-deficient mice [70]. Ac-hE18A-NH_2_ also decreases total cholesterol and lipid peroxide levels and increases HDL-associated PON activity in the plasma of Watanabe hypercholesterolemic rabbits [71]. A modified form of this peptide, defined as AT1-5261, which contains less acidic residues in the non-polar face and more acidic residues in the polar face, displays improved cholesterol efflux capacity, that is further enhanced when AT1-5261 was complexed with phosphatidylcholine. The intervention with AT1-5261 efficiently reduces high-fat diet-induced atherosclerosis when administered for six consecutive weeks by intraperitoneal injection to either LDLR or apoE-deficient mice [25].

Oral administration of mR18L, the cationic single domain amphipathic peptide, reduces cholesterol plasma levels and atherosclerotic lesion in ApoE-deficient mice [72]. In LDLR-deficient mice, mR18L chronic treatment decreases plasma cholesterol to similar levels than Ac-hcE18-NH2, but inhibits the atherosclerotic lesion to a lower extent [28]. mR18L also showed anti-inflammatory properties by inhibiting lipopolysaccharides (LPS) effect in endotoxemic rats [73].

The EpK peptide, which contains an LDLR-binding region and a lipid-binding region, binds to HDL and protects macrophages against LPS-induced inflammation. This peptide also promotes cholesterol efflux with higher efficiency than ApoA-I and ApoE3 [26]. In addition, the lentivirus-mediated hepatic expression and secretion of EpK strongly reduced the atherosclerotic plaque in ApoE-deficient mice despite its limited effect on plasma lipid levels [74]. hEp not only reduces plasma cholesterol levels but also reduces the progression of atherosclerosis in aged female apoE-deficient mice with existing aortic lesions, suggesting that hEp may be a promising anti-atherosclerotic therapy [75].

### 3.4. ApoJ-Based Peptides

ApoJ expression increases under stress and forms part of the control system against protein unfolding. ApoJ is involved in apoptosis, cell adhesion, tissue remodeling, immune system regulation, and oxidative stress [76]. As a consequence of its ubiquitous expression and multiple effects, ApoJ has been involved in several pathological processes, including aging, cancer, diabetes, kidney disease, Alzheimer’s disease, and atherosclerosis. ApoJ has been detected in the intima and media in early atherosclerosis lesions [77]. The ApoJ/PON1 ratio is increased in patients at high risk of cardiovascular disease [78] and in mouse models of atherosclerosis [79]. ApoJ exerts anti-atherosclerotic effects, mainly the induction of RCT [79]. Oral administration of D-(113–122) ApoJ-based peptides was shown to be as effective as ApoJ protein in inhibiting atherosclerosis in ApoE-deficient mice [31]. After acute administration of D-(113–122) ApoJ peptide in this model, the anti-inflammatory properties of HDL improved, as well as the plasma cholesterol efflux capacity. Incubation of ApoJ with plasma from these mice reduces lipid peroxides and increases PON activity [31]. Similarly, ApoJ peptide increases PON and HDL anti-inflammatory properties while decreases lipid peroxide levels in the plasma of monkeys [31]. Moreover, ApoJ shares other atheroprotective effects with HDL, such as the induction of cytoprotective and anti-inflammatory actions [80]. D-(113–122) ApoJ peptide improves the functionality of lipoproteins in LDLR-deficient mice fed an atherogenic diet [81]. In particular, LDL particles become resistant to aggregation and with lower electronegativity, whereas HDL shows increased antioxidant capacity and ability to promote cholesterol. These results suggest the possibility of using this apoJ-based peptide as a therapeutic tool for atherosclerosis.

### 3.5. LRP1-Based Peptides

Smooth muscle cells are the main cellular components of the vascular wall, and up to 40% of foam cells previously identified as monocyte-derived macrophages in human atherosclerosis are SMCs [82,83]. One of the main mechanisms for SMC-foam cell formation is the uptake of proatherogenic extracellular matrix-retained agLDL through the LRP1 receptor [53,54,55,56,57,58]. Both the susceptibility of LDL particles to aggregate [84] and the circulating levels of LRP1 [85] are predictive of cardiovascular risk, independently of traditional risk factors. LRP1 is overexpressed in human lipid-enriched advanced coronary atherosclerotic plaques [86]. In addition, clinical studies have suggested that different genetic variants of the receptor are independent risk factors for cardiovascular conditions, including CAD [87,88]. These findings suggest a key role of foam-SMC in atherosclerosis progression and open new strategies to modulate atherosclerosis. The new family of LRP1-based peptides efficiently inhibits the proatherogenicity of LDL and its capacity to generate foam SMC [57,58]. In fact, a strong correlation between the inhibitory activity of peptides on SMase-induced LDL aggregation and hcVSMC-cholesterol loading has been reported using biochemical and cell-based standardized assays (58). These studies highlight the potential relevance and clinical interest of these compounds in the treatment of atherosclerosis and the management of cardiovascular diseases.

## 4. Peptide-Based Immunization against Atherosclerosis

Several studies have reported the efficacy of newly developed peptides (Table 2) to induce protective immune response (specific antibodies) against atherosclerosis.

### 4.1. ApoA-I-Based Peptides

The recognition of modified oxidized LDL (oxLDL) by the immune system is a key step in the activation and regulation of the inflammatory process that occurs during atherosclerosis [96]. Anti-oxLDL antibody production induced by oxLDL-based peptide immunization confers atheroprotection in animal models [97,98]. Wool et al. determined the effects of peptides 4F and Propeptide (a tandem of two α-helices of 4F separated by proline) on antibody titers against specific epitopes of oxLDL in two different stages of atherosclerotic plaque progression in ApoE^−/−^ mice [89]. Immunization with both peptides increases the production of antibodies, including the natural IgM E06/T15 antibodies, that recognize oxidized phospholipids. The mechanism by which these peptides increase natural antibodies is still unknown. Immunization with 4F peptide decreases atherosclerosis in early but not in advanced stages.

### 4.2. ApoB-100-Based Peptides

ApoB-100-based peptides show pro-inflammatory properties such as activation of T cells, B cells, and monocyte-macrophage system. Therefore, therapies based on immunization with these peptides have been developed [90].

Immunization with p210 peptide (3136 to 3155 residues of human apoB-100) has been shown to decrease dendritic cell and macrophage infiltration, and reduce atherosclerotic plaque burden in ApoE^−/−^ mice [91,92]. Peptide-induced atheroprotective effects include CD8^+^ T-cell response activation [92].

Antibodies against LDL(-) have been detected in the blood and atherosclerotic plaques of rabbit models and humans [99,100]. These antibodies were reported to be atheroprotective in mice [101]. An antibody named 2C7 was used to design a peptide (p2C7) based on one particular epitope that induces macrophage inflammation [102]. The authors suggested that immunization with this or other putative LDL(-) mimetic peptides may be a promising strategy for developing vaccines against atherosclerosis [102].

### 4.3. LRP1-Based Peptides

LRP1 interacts with several anti-inflammatory ligands including ApoE that blocks interleukin-1 receptor associated kinase-1 (IRAK-1) activation, helping to reduce the inflammatory effects of nuclear factor-kappa B (NF-κB) in VSMCs [103]. LRP1 contributes to the protective effect of transforming growth factor-β (TGF-β) on macrophage Wnt5a signaling [104]. LRP1 also mediates the anti-inflammatory effects of protease-inhibitor complexes [105,106,107]. These anti-inflammatory mediators interact with sequences of the α-chain that are outside of the CR9 domain, where P3, the sequence involved in the interaction with agLDL, is located. This allowed the development of CR9-based peptide therapeutic strategies specifically focused on counteracting the pathological interaction of LRP1 with agLDL, without altering LRP1 anti-inflammatory signaling. Polyclonal antibodies generated against P3 sequence (involved in binding to aggregated LDL) efficiently prevented the formation of foam cells from hcVSMCs. Moreover, anti-P3 Abs efficiently prevented agLDL-induced LRP1 upregulation and counteracted the down-regulatory effect of agLDL on hcVSMC migration [57]. P3 immunization raised the production of specific anti-P3 antibodies that drastically reduce the accumulation of cholesteryl esters and the levels of pro-inflammatory markers in the aorta of rabbits [95]. These studies showed that domain CR9 is critical for LRP1-mediated agLDL binding and internalization in hcVSMCs and opens a new avenue for an innovative strategy in the treatment of local vascular lipid deposition in atherosclerosis.

## 5. Peptides in Clinical Phases

Several peptides are currently undergoing clinical evaluation. As shown in Table 3, most of them are based on ApoA-I sequences or ApoA-I sequences complexes with lipids. It should be noted that only one has already reached phase 3, while the rest are in phase 1 or 2.

### 5.1. ApoA-I-Based Peptides

Among all ApoA-I-derived peptides, few have succeeded in clinical studies. The oral administration of D-4F reduced HDL inflammation index (HII) in high-risk cardiovascular patients [108]. HII was significantly reduced with 300 mg dose at 4 h of administration, and with 500 mg dose at 2 h. However, D-4F bioavailability was below 1%. The results of this first study conclude that oral D-4F is safe, well tolerated, and efficient to increase HDL anti-inflammatory index. In the first multi-dose study, motivated by the above results, oral D-4F pharmacokinetics and pharmacodynamics were evaluated in statin-treated patients with CAD. Eight hours after oral administration, HII was reduced by 28% (range, 1.25–0.86) in the placebo, whereas in the D-4F groups the reduction was by 55% (1.35–0.57) at 300 mg, and by 49% (1.22–0.63) at 500 mg. However, HII returned to baseline values after 24 h of peptide administration, suggesting that prolonged anti-inflammatory effects require multiple or higher daily doses. This study demonstrated that statin combined with D-4F significantly decreases HII and increases HDL anti-inflammatory efficacy to a higher extent than statin monotherapy.

In other clinical study, L-4F peptide was administered either via intravenous infusion over 7 days or via subcutaneous injection over 28 days to patients with CAD [109]. Blood peptide levels were sufficient to cause anti-atherosclerotic effects in animal models and anti-inflammatory effects in humans (D-4F results in phase 1). However, L-4F treatment did not improve HDL functionality, suggesting that in vivo anti-atherosclerotic effects of ApoA-I mimetic peptides were independent of their in vitro properties. This fact caused some controversy about the potential clinical use of ApoA-I mimetic peptides.

The peptide 5A combined with sphingomyelin, called Fx-5A, is currently undergoing a phase 1A clinical trial [115]. This peptide was designed to eliminate excessive intracellular cholesterol via the ABCA1 transporter. Fx-5A peptide efficiently blocks the progression of atherosclerosis, thereby promoting cholesterol release and inflammation reduction in healthy volunteers [110].

Several molecules mimicking HDL/ApoA-I have been developed to enhance RCT. Two of these molecules, ApoA-I Milano and CSL-112, have been already tested in clinical trials. ApoA-I Milano peptide was administered intravenously preparing a peptid/lipid complex (ETC-216) and its efficacy was assessed through the evaluation of atherosclerotic plaque burden by means of intravascular ultrasound (IVUS) [116]. This peptide efficiently reduced the plaque burden in patients with acute coronary syndrome (ACS); however, it did not produce a regression of advanced atherosclerotic lesions in patients with ACS in another randomized clinical trial (project MDCO-216) [111]. Finally, the parent company decided to end this project. CSL-112 has proven to increase pre-ßHDL and RCT in healthy individuals and administered doses were well tolerated and without evidence of toxicity [117]. The safety, tolerability, pharmacokinetics, and pharmacodynamics of CSL-112 were assessed in a phase 2b clinical trial (AEGIS-I) that included patients who had recently suffered an acute myocardial infarction [118]. The results of phase 3 will be available soon (AEGIS-II) [112].

Another ApoA-I mimetic complexed with phospholipids, CER-001, was investigated in patients with a history of percutaneous coronary intervention (PCI) or with at least 20% coronary artery luminal narrowing. This phase 2 study revealed that CER-001 did not reduce coronary atherosclerosis, as assessed by IVUS and quantitative coronary angiography (QCA) [113]. In another study with healthy volunteers, CER-001 caused elevations in plasma cholesterol, as well as total and free cholesterol in the HDL fraction, suggesting an increase in RCT [119].

### 5.2. ApoE-Based Peptides

The Ac-hE18A-NH_2_ peptide, under the commercial name AEM-28, demonstrated its safety profile in phase 1a and phase 1b/2a clinical trials. Intravenous administration to 51 patients resulted in a reduction of over 50% in TGs and VLDL-C compared to placebo treatment [114].

## 6. Opportunities and Risk of Peptide-Based Treatments

The development of liver X receptor (LXR) agonists and microsomal triglyceride transfer protein (MTP1) drugs as emergent therapies for atherosclerosis were a big challenge in the last years. LXR are essential nuclear receptors that exert atheroprotective properties due to their capacity to decrease intestinal cholesterol absorption and induce RCT. Therefore, a big effort was invested to develop efficient LXR agonists [120]. MTP is an enzyme responsible for the transfer of TGs to apolipoprotein B in enterocytes and hepatocytes to configure VLDL and LDL, respectively. MTP inhibitors were designed to reduce the secretion of TG-enriched lipoproteins and hypertriglyceridemia. Both LXR agonists and MTP inhibitors hold crucial anti-atherosclerotic properties [121,122] but unfortunately induce serious gastrointestinal and hepatic adverse effects [123,124], which discourage their use. In this scenario, the new apolipoprotein-based peptides emerge as a new opportunity to modulate lipid, lipoprotein profile, and inflammatory processes with lower toxic adverse effects since they exert their main effects through outside cell. Most of the problems potentially associated with apolipoprotein-based peptides can be solved by using specific routes of administration. Peptide complexes for administration include nanoparticles, liposomes, hydrogels, and transdermal delivery systems among others [125,126]. Most of the apolipoprotein-based peptides reaching clinical phases have been stabilized in lipid complexes or used in stabilized retro-enantio peptide versions and administered by intravenous or subcutaneous injections. The main advantages and risks of using apolipoprotein-based peptides have been summarized in Table 4.

## 7. Conclusions and Further Directions

This is a comprehensive review focused on the use of peptides as anti-atherosclerotic tools. It integrates results from in vitro and in vivo studies, and clinical trials to summarize the potential applicability of these novel therapeutic tools. We hope to arise the interest of researchers and clinicians involved in the Atherosclerosis field. As shown in Figure 1, mimetic peptides exert a wide variety of beneficial effects, both extra- and intracellular. These effects include improvement of lipid and lipoprotein profiles and protection of LDL against modification. At the intracellular level, peptides exert anti-inflammatory, antioxidant, and anti-foam cell formation effects that result in atherosclerosis reduction. Mimetic peptides are promising therapeutic tools to target atherosclerosis because they cause low cytotoxicity and have low associated production costs; however, they still have to face many challenges in drug development and clinical trials.

## 8. Review Methodology, Search Strategy and Selection Criteria

We performed an extensive (systematic) search in PubMed by combining several terms, mainly: mimetic peptides, apolipoprotein, LRP1, atherosclerosis, LDL, and HDL. Relevant articles were chosen and their references were secondarily searched for additional relevant articles with no limit on the original date of the article.

## Figures and Tables

**Figure 1 jcm-10-03571-f001:**
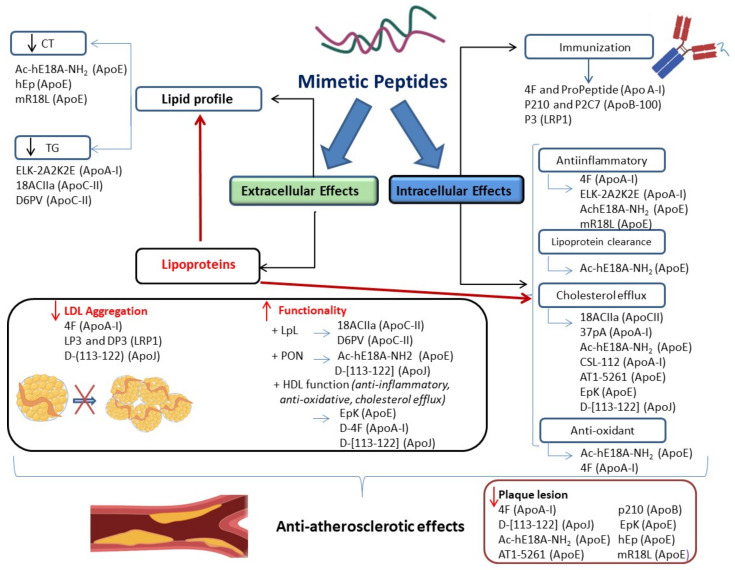
Schematic representation of the main extracellular and intracellular protective functions of anti-atherosclerotic peptides either used as peptidomimetics or as peptides raising immunization.

**Table 1 jcm-10-03571-t001:** Apolipoprotein and LRP1-based peptides sequences.

Derived	Peptide	Sequences	Ref
ApoA-I	18A	Ac-DWLKAFYDKVAEKLKEAF- NH_2_	[18]
	37pA	18A-P-18A	[19]
	4F (L-4F and D-4F)	Ac-DWFKAFYDKVAEKFKEAF-NH_2_	[20]
	ELK-2A2K2E	EKLKAKLEELKAKLEELL-P-EKLKAKLEELKAKLEELL	[21]
ApoC-II	18A-C-II-a	18A-P-AMSTYTGIFTDQVLSVLKGEE	[22]
	D6PV	DYLKEVFEKLRDLYEKFTPAVSTYTGIFTDQVLSVLKGEE	[23]
ApoE	Ac-hE18A-NH_2_	Ac-LRKLRKRLLR-18A-NH_2_	[24]
	AT1-5261	EVRSKLEEWFAAFREFAEEFLARLKS	[25]
	EpK	NH_2_-CRRKLRKRLLRKKKKKKQVAEVRAKLEEQAQQIRLQAE-COOH	[26]
	hEp	EELRVRLASHLRKLRKRLLRDADDLQKRLAVYEEQAQQIRLQAEAFQARLKSWFEPLVEDM	[27]
	mR18L	AcGFRRFLGSWARIYRAFVGNH2	[28]
	ApoE (141–155) dimer	Ac-LRKLRKRLLRDADDLLRKLRKRLLRDADDL	[29]
	ApoEdp	Ac-LRKLRKRLLLRKLRKRLL-NH_2_	[30]
ApoJ	D-(113–122) ApoJ	Ac-LVGRQLEEFL-NH_2_	[31]
LRP1	LP3	H-GDNDSEDNSDEENC-NH_2_	[32]
	DP3	H-NEEDSNDESDNDG-NH_2_	[32]

**Table 2 jcm-10-03571-t002:** Apolipoprotein and LRP1-based peptides used for immunization experiments.

Original	Peptide	Sequences	Animal Model	Adjuvant/ Carrier	Administration Route	Ref
ApoA-I	4F	DWFKAFYDKVAEKFKEAF	apoE ^−/−^ mice	sterile PBS	Intraperitoneal injections	[89]
	Pro peptide	4F-P-4F	apoE ^−/−^ mice	sterile PBS	Intraperitoneal injections	[89]
ApoB-100	p210	KTTKQSFDLSVKAQYKKNKH	apoE ^−/−^ mice	Cholera toxin B or Alum + cationized BSA	Intranasal or subcutaneous injection	[90,91,92]
			LDLR ^−/−^/hapoB-100 mice	Alum + cationized BSA	Subcutaneous injection	[93]
	p2C7	CMPSVILPSC	LDLR ^−/−^ mice	none	Passive immunization	[94]
LRP1	P3	GDNDSEDNSDEENC	NZW Rabbit	KLH (Keyhole limpet haemocyanin)	Subcutaneous injection	[95]

**Table 3 jcm-10-03571-t003:** Apolipoprotein and HDL-mimetic peptides in clinical trials.

Original	Peptide	Company	Administration Route	Conclusions	Stage	Ref and NCT
ApoA-I	D-4F	Novartis	Oral	Poor bioavailability HII improve	Phase 2	[108]
	L-4F	Novartis	Intravenous or subcutaneous	Reach plasma levels HII not improve	Phase 2	[109]
	5A	NHLBI	Intravenous	Not yet	Phase 1	[110] NCT04216342
HDL mimetic	ApoA-I milano	Esperion Therapeutics Pfizer’s	Intravenous	No plaque regression	Phase 1	[111] NCT02678923
	CSL-112	CSL Behring Inc	Intravenous	Not yet	Phase 3	[112] NCT03473223
	CER-001	Cerenis Therapeutics	Intravenous	No plaque reduction	Phase 2	[113] NCT01201837
ApoE	Ac-hE18A-NH_2_	LipimetiX Development	Intravenous	Safety profile Reduce TG and VLDL-C	Phase 1	[114] NCT02100839

**Table 4 jcm-10-03571-t004:** Opportunities and risks associated with apolipoprotein-and LRP1-based peptides for therapy.

Opportunities	Risks
Improvement of lipid and lipoprotein profile and LDL protection	Interactions derived from the multiple functions of apolipoproteins
Wide versatility to treat CVDs due to low product costs	Conformational peptide alterations depending on pH
New knowledge about non-classical roles of apolipoproteins and their implications in multiple events	Requirements of controlled release delivery device or frequent dosing
Extracellular and focalized actions	Ability to overcome cell membrane permeability
Wide variety of routes for peptide delivery	Low oral bioavailability

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
