# Peer review of "Apolipoprotein and LRP1-Based Peptides as New Therapeutic Tools in Atherosclerosis"

_jcm, 2021, doi:10.3390/jcm10163571_

Round 1

Reviewer 1 Report

The authors have responded appropriately to the critique of this reviewer and the paper meets standards for publication. However, there are a lot of spelling errors and grammatical problems that need to be corrected during copyediting.

Here are a few examples 

Line 155 "or LDL aggregation degree" should be "degree of LDL aggregation"

Typo on line 299 "properties than render" should say "properties that render"

Line 512 awkward phrasing "which force to minimize their use" could say "which discourage their use"

Author Response

Reviewer 1:

The authors have responded appropriately to the critique of this reviewer and the paper meets standards for publication. However, there are a lot of spelling errors and grammatical problems that need to be corrected during copyediting.

Following your recommendation, we have exhaustively revised the English grammar through all the text

Here are a few examples 

Line 155 "or LDL aggregation degree" should be "degree of LDL aggregation". We have replaced this sentence as the Reviewer indicates

Typo on line 299 "properties than render" should say "properties that render". We have replaced this text in order to clarify the meaning of the sentence.

Line 512 awkward phrasing "which force to minimize their use" could say "which discourage their use". We have changed the sentence as indicated by the Reviewer

Reviewer 2 Report

The purpose of the article by Benitez-Amaro et al. is to present the current state of knowledge regarding the importance of apolipoprotein- and LRP1-based peptides as new therapeutic tools in atherosclerosis.

In a previous review of the article, I suggested that the article needs to be supplemented:
- In the introduction, the epidemiology of atherosclerosis and its social importance must be discussed.
- An answer to the question of how literature was collected for review must be included. Ideally in the chapter "review methodology".
- At the end of the review, I would also add an illustration, graph or table about the opportunities and risks associated with this treatment option. It would be an interesting critical summary of the topic.

In the current version of the article, all the above suggestions have been fully taken into account.
I have no further comments.

Author Response

In the current version of the article, all the above suggestions have been fully taken into account. I have no further comments.

Thanks for your positive answer

This manuscript is a resubmission of an earlier submission. The following is a list of the peer review reports and author responses from that submission.

Round 1

Reviewer 1 Report

This is a very well written overview of a potential new target for the treatment of atherosclerosis. You need to complete the following aspects:

1. In the introduction, the epidemiology of atherosclerosis and its social importance must be discussed.

2. An answer to the question of how literature was collected for review must be included. Ideally in the chapter "review methodology".

3. At the end of the review, I would also add an illustration, graph or table about the opportunities and risks associated with this treatment option. It would be an interesting critical summary of the topic. 

Reviewer 2 Report

This is a nicely written and organized overview of the topic. Good tables and figure. Needs copyediting for language. A few additions would improve the manuscript:

1) Elaborate further on ApoE mimetics such as mR18L and EpK.

2) Discuss problems and toxicities in context of severe problems with LXR agonists and MTP1 drugs.

3) Address drug delivery systems - a lot of cutting edge work being done.

Minor issues:

Please correct: "cholestesteryl ester" in Introduction. ApoA-I not consistent, sometimes ""ApoA-I" and other times "apoA-I". High-density lipoproteins  (HDL) repeated multiple times. Can just use abbreviation after first use.

On page 6 "In this section, we will revise the potential mechanisms" is incorrect. I believe it should read "In this section, we will review the potential mechanisms"

On page 8 "In line, D-" What does "In line" mean?

CAD abbreviation used for coronary artery disease on page 8, so abbreviation can be used again on page 11 (twice).